# Characteristics and Clinical Ocular Manifestations in Patients with Acute Corneal Graft Rejection after Receiving the COVID-19 Vaccine: A Systematic Review

**DOI:** 10.3390/jcm11154500

**Published:** 2022-08-02

**Authors:** Kenta Fujio, Jaemyoung Sung, Satoru Nakatani, Kazuko Yamamoto, Masao Iwagami, Keiichi Fujimoto, Hurramhon Shokirova, Yuichi Okumura, Yasutsugu Akasaki, Ken Nagino, Akie Midorikawa-Inomata, Kunihiko Hirosawa, Maria Miura, Tianxiang Huang, Yuki Morooka, Mizu Kuwahara, Akira Murakami, Takenori Inomata

**Affiliations:** 1Department of Ophthalmology, Juntendo University Graduate School of Medicine, Tokyo 113-0033, Japan; k.fujio.zz@juntendo.ac.jp (K.F.); jsung1@usf.edu (J.S.); satoru-n@juntendo.ac.jp (S.N.); k-fujimoto@juntendo.ac.jp (K.F.); h-shokirova@juntendo.ac.jp (H.S.); y-okumura@juntendo.ac.jp (Y.O.); y-akasaki@juntendo.ac.jp (Y.A.); k-hirosawa@juntendo.ac.jp (K.H.); maria-k@juntendo.ac.jp (M.M.); h.tianxiang.zb@juntendo.ac.jp (T.H.); y.morooka.df@juntendo.ac.jp (Y.M.); mz-ohno@juntendo.ac.jp (M.K.); amurak@juntendo.ac.jp (A.M.); 2Department of Digital Medicine, Juntendo University Graduate School of Medicine, Tokyo 113-0033, Japan; k-nagino@juntendo.ac.jp; 3Morsani College of Medicine, University of South Florida, Tampa, FL 33612, USA; 4Department of Respiratory Medicine, Nagasaki University Hospital, Nagasaki 852-8102, Japan; kazukomd@nagasaki-u.ac.jp; 5Department of Health Services Research, Faculty of Medicine, University of Tsukuba, Tsukuba 305-8575, Japan; iwagami-tky@umin.ac.jp; 6Department of Hospital Administration, Juntendo University Graduate School of Medicine, Tokyo 113-0033, Japan; ak-inomata@juntendo.ac.jp; 7AI Incubation Farm, Juntendo University Graduate School of Medicine, Tokyo 113-0033, Japan

**Keywords:** coronavirus disease 2019 (COVID-19), SARS-CoV-2, vaccine, corneal transplantation, corneal graft, immune response, tolerance, corneal graft rejection, systematic review

## Abstract

This study aimed to determine the characteristics and clinical ocular manifestations of acute corneal graft rejection after coronavirus disease 2019 (COVID-19) vaccination. We conducted an online search of the PubMed and EMBASE databases. Data on recipients’ characteristics, corneal transplantation types, interval between vaccination and allograft rejection, clinical manifestations, and graft rejection medication were extracted. Thirteen articles on 21 patients (23 eyes) with acute corneal graft rejection after COVID-19 vaccination, published between April and December 2021, were included. The median (interquartile range) age at the onset of rejection was 68 (27–83) years. Types of transplantation included penetrating keratoplasty (12 eyes), Descemet membrane endothelial keratoplasty (six eyes), Descemet stripping automated endothelial keratoplasty (four eyes), and living-related conjunctival-limbal allograft (one eye). The interval between vaccination and rejection ranged from 1 day to 6 weeks. Corneal edema was the leading clinical manifestation (20 eyes), followed by keratic precipitates (14 eyes) and conjunctival or ciliary injection (14 eyes). Medications included frequently applied topical corticosteroids (12 eyes), followed by a combination of topical and oral corticosteroids (four eyes). In addition, the clinical characteristics of corneal allograft rejection after COVID-19 vaccination were identified. Corneal transplant recipients may require further vaccination, necessitating appropriate management and treatment.

## 1. Introduction

Severe acute respiratory syndrome coronavirus 2 (SARS-CoV-2) caused an unprecedented worldwide pandemic of the coronavirus disease 2019 (COVID-19) [1,2]. On 10 March 2022, the total number of reported COVID-19 cases surpassed 450 million [3]. Various complications have been reported after COVID-19 infection, including thrombosis [4], neuropathy [5], and ocular complications such as conjunctivitis [2], uveitis [6], and optic neuritis [7]. In December 2020, a large-scale vaccination program was initiated in Israel; this was later adopted worldwide to contain the spread of SARS-CoV-2 infection. By the end of 2021, nearly half of the world’s population had completed the vaccination schedule according to the recommendations of the manufacturers of the various vaccine brands that were used [8]. However, with the emergence of variants of the virus, the US Food and Drug Administration and European Medicines Agency approved additional booster schedules for five COVID-19 vaccines by the end of March 2022 [9]. With immense-scale vaccination being performed, post-vaccination complications, including hypercoagulability, Guillain–Barre syndrome, and myocarditis, are continuously being reported and investigated [10]. Additionally, allograft organ rejection has also been reported [11], which warrants close monitoring and measures being undertaken by clinicians to prevent the occurrence of these side effects in high-risk populations [12,13,14,15].

Corneal allograft transplantation is the most frequently performed organ transplant procedure worldwide [16,17]. The cornea is one of the few organs with an immune privilege and, thus, a decreased rate of allograft rejection [18,19]. However, in April 2021, two cases of acute corneal rejection after Descemet membrane endothelial keratoplasty (DMEK) with a possible association with COVID-19 vaccination were reported [20]. In addition, several reports have previously described corneal allograft rejection being triggered by vaccinations, including those secondary to Influenza and Hepatitis B vaccine administration [21]. These reports have raised concerns among ophthalmologists as future booster doses of COVID-19 vaccines are being discussed, which may impact outcomes in future corneal allograft recipients [22,23]. Thus, we must recognize the clinical features, risk factors, and course of allograft rejection, as well as currently known data on post-rejection rescue attempts. This information will help to establish effective care and preventive measures against allograft rejection in future corneal transplant recipients with a history of recent COVID-19 vaccination or for those planning COVID-19 vaccination.

We performed a systematic review of the corneal allograft rejection cases reported after COVID-19 vaccination to identify the novel recipient characteristics and clinical findings of allograft rejection. To our knowledge, this is the first systematic review of corneal allograft rejections after COVID-19 vaccine administration.

## 2. Materials and Methods

### 2.1. Outcomes

The primary aim of this study was to systematically evaluate and characterize the currently reported cases of acute corneal graft rejection after COVID-19 vaccine administration. In particular, we focused on variables such as age, sex, and ethnicity of patients; type of vaccine; the number of vaccinations; type of corneal transplantation; the interval between corneal transplantation and rejection; ocular findings; the interval between vaccination and rejection; and medications used to treat rejection.

### 2.2. Search Strategy

An extensive search strategy was designed to retrieve all articles published by 23 February 2022, combining generic terms—“(coronavirus 2019)” OR “COVID-19” OR “COVID” OR “SARS-CoV-2” OR “(2019 novel coronavirus)” OR “2019-nCoV)” AND “((cornea) OR (corneal))” AND “(rejection)”—in key electronic bibliographic databases (PubMed and EMBASE). In addition, we followed the Preferred Reporting Items for Systematic Review and Meta-Analyses (PRISMA) guidelines [24]. The inclusion and exclusion criteria are detailed in Table 1.

Search results were compiled using Endnote software X9.3.3 (Clarivate Analytics, Philadelphia, PA, USA). Two independent researchers (K.F. and T.I.) screened the retrieved articles in accordance with the defined quality standards for reporting systematic reviews and meta-analyses for observational studies [25]. Additionally, the same investigators independently assessed the full texts of eligible articles to reach a consensus.

### 2.3. Risk of Bias Assessment

The risk of bias in the individual studies was assessed using either the Joanna Briggs Institute (JBI) Critical Appraisal Checklist for Case Reports or the JBI Critical Appraisal Checklist for Case Series [26]. The checklists for case reports and case series consist of 8 and 10 items, respectively, with response choices of “yes”, “no”, “unclear”, or “not applicable”. Two investigators (K.F. and K.N.) independently assigned an overall risk of bias to each eligible study, and if they disagreed, a third reviewer (T.I.) was consulted. The risk of bias was determined considering the total number of “yes” responses, with ≥70%, 50–69%, and ≥49% of the responses indicating low, moderate, and high risk of bias, respectively [27].

### 2.4. Data Extraction

Two independent reviewers (K.F. and T.I.) extracted data from each eligible article using a standardized data extraction sheet and then cross-checked the results. Disagreements between the reviewers regarding extracted data were resolved through discussion with a third reviewer (J.S.). The following information was extracted: first author’s name; date of publication; type of study (case report and case series); country; characteristics of patients with acute corneal graft rejection after COVID-19 vaccination, including age, sex, and race; type of vaccine; the number of vaccinations; type of corneal transplantation; the interval between corneal transplantation and rejection, the interval between vaccination and rejection; ocular findings; and medications.

### 2.5. Statistical Analysis

Data analyses were performed considering the Updated Method Guidelines for Systematic Reviews in the Cochrane Collaboration Back Review Group [28].

The mean (±standard deviation) or median (interquartile range) interval between corneal transplantation and rejection and the interval between vaccination and rejection were analyzed.

## 3. Results

Figure 1 illustrates the screening process. Twenty-four articles were identified through the database search on 13 February 2022 [20,29,30,31,32,33,34,35,36,37,38,39,40,41,42,43,44,45,46,47,48,49,50,51]. After reviewing the titles and abstracts of the 24 articles, 10 were excluded based on the article type (letter to the editor and review) [44,47], corneal graft rejection after COVID-19 [45,46,48,50], eye banking issues during the COVID-19 pandemic [42,49], and other unrelated topics [43,51]. Fourteen articles were selected for full-text screening, and one was excluded because of the lack of clinical history [20,29,30,31,32,33,34,35,36,37,38,39,40,41]. Finally, 13 articles [20,29,30,31,32,33,34,35,36,37,38,39,40] met the inclusion criteria and were included in the systematic review. The results of the JBI Critical Appraisal Checklists for case reports and case series are summarized in Table 2. The 13 articles included showed a low or moderate risk of bias.

### 3.1. Study Characteristics and Demographic Features

The articles included in this systematic review were published between 29 April 2021, and 23 December 2021 [20,29,30,31,32,33,34,35,36,37,38,39,40]. Nine articles were case reports [30,31,32,34,35,37,38,39,40], and four were case series [20,29,33,36]. Four articles were from the United States of America [32,33,34,35]; two each from the United Kingdom [20,31] and India [37,40], and one each from Brazil [39], Greece [36], Israel [29], Lebanon [30], and Italy [38]. A total of 21 patients (23 eyes) who experienced corneal graft rejection after COVID-19 vaccine administration were identified in the 13 studies (Table 3). Thirteen articles reported the age (median [interquartile range]: 68 [27–83] years, *n* = 21) at the onset of acute corneal graft rejection. Thirteen articles described the sex of the patients (10 males and 11 females) [20,29,30,31,32,33,34,35,36,37,38,39,40], and four articles mentioned the race of the patients (10 Caucasians and one Black) [20,33,36,38]

### 3.2. Types of Vaccines and the Number of Vaccinations

Thirteen articles reported four types of vaccines that were administered, including BNT162b2 (Pfizer/BioNTech, Mainz, Germany; eight [38.1%] cases) [20,29,30,31,32,38], mRNA-1273 (Moderna, Cambridge, MA, USA; eight [38.1%] cases), [33,34,35,36] ChAdOx1 (University of Oxford/AstraZeneca, Oxford, UK; four [19.0%] cases) [36,37,40], and CoronaVac (Sinovac Biotech, Beijing, China; one [4.8%] case) [39]. Approximately 66.7% of these patients received the first vaccine dose, whereas 33.3% received the second vaccine dose.

### 3.3. Types of Corneal Transplantation and the Interval between Corneal Transplantation and Rejection

Thirteen articles reported 12 cases of penetrating keratoplasty (PKP, 52.2%, 12/23 eyes) [29,31,33,35,36,37,38,39,40], six cases of DMEK (26.1%, 6/23 eyes) [20,30,33,36], four cases of Descemet stripping automated endothelial keratoplasty (DSAEK, 17.4%, 4/23 eyes) [32,33,36], and one case of living-related conjunctival-limbal allograft (LR-CLAL, 4.3%, 1/23 eyes) [34]. Among these cases, the interval between corneal transplantation and rejection ranged from 14 days [20] to 25 years [38] (median: 2 years [20,29,30,31,32,33,34,35,36,37,38,39,40]).

### 3.4. Interval between Vaccination and Rejection, Clinical Ocular Manifestations, and Medications

The interval between COVID-19 vaccination and rejection ranged from 1 day [39] to 6 weeks [40] (mean, 10.4 days; median, 7 days [20,29,30,31,32,33,34,35,36,37,38,39,40]). The clinical ocular findings are shown in Table 3 and Table 4. Corneal edema was the main clinical ocular manifestation (87.0%, 20/23 eyes), followed by keratic precipitates (60.9%, 14/23 eyes), conjunctival or ciliary injection (60.9%, 14/23 eyes), inflammatory reaction in the anterior chamber (43.5%, 10/23 eyes), Descemet membrane folds (26.1%, 6/23 eyes), corneal endothelial rejection line (13.0%, 3/23 eyes), and fluid at the laser in situ keratomileusis (LASIK) interface (4.3%, 1/23 eyes).

Thirteen articles reported medications (Table 5) for corneal graft rejection, including (1) frequently applied topical corticosteroids (52.1%, 12/23 eyes); (2) a combination of topical and oral corticosteroids (13.0%, 4/23 eyes); (3) a combination of topical and intravenous corticosteroids (4.3%, 1/23 eyes); (4) a combination of topical corticosteroids and subconjunctival or intracameral corticosteroid injections (8.7%, 2/23 eyes); (5) a combination of topical and oral corticosteroids and subconjunctival corticosteroid injections (4.3%, 1/23 eyes); (6) a combination of oral and topical corticosteroids and immunosuppressants (4.3%, 1/23 eyes); (7) a combination of topical corticosteroids and vitamin D supplements (4.3%, 1/23 eyes). Finally, nine eyes (39.1%, 9/23 eyes) developed corneal graft failure after vaccination.

## 4. Discussion

Since the beginning of the COVID-19 pandemic in 2019, efforts toward vaccination have continued worldwide due to the uncontrollable spread of SARS-CoV-2 infection and the increasing number of COVID-19 cases. However, despite the low occurrence rates of high-risk complications after vaccination, an appreciable number of vaccine recipients have experienced a wide range of post-vaccination symptoms. At present, COVID-19 vaccines are believed to be associated with a spectrum of systemic symptoms, and appropriate interventions should be undertaken on a case-by-case basis. In this systematic review, we extracted data regarding clinical features of acute corneal allograft rejection (21 patients [23 eyes]) that occurred after the administration of the COVID-19 vaccine. Among them, >95% of eyes (22/23 eyes) had corneal allograft rejection within 3 weeks from vaccination (mean: 10.4 days, median: 7 days). As the global society promotes additional booster schedules in consideration of emerging variants, it is empirical that the effects of these vaccines on corneal grafts be elucidated. To establish appropriate immune-modulatory interventions, continued data accrual and investigation of the effects of COVID-19 vaccines should be conducted by observing the vaccination course and associated physiological changes in corneal allograft recipients.

In our analysis of 21 patients (23 eyes) with signs of corneal allograft rejection after receiving COVID-19 vaccines, the median interval between corneal transplantation and graft rejection after COVID-19 vaccination was 2 years. Two of these patients (two eyes) underwent PKP > 20 years ago and had no history of acute or chronic corneal allograft rejection [33,38]. The long-term stability and temporariness of vaccination and allograft rejection suggest that immune responses to the vaccine may have played a role in transplant rejection. Two separate studies investigated the possibility of confounding viral infections, including herpes simplex and varicella-zoster viral infections, which are known risk factors for corneal allograft rejection after PKP or DMEK [20,38], through anterior-chamber, aqueous-humor polymerase chain reaction (PCR). However, PCR results for confounding viral infections were negative in both patients. Additionally, two cases (four eyes) of post-DMEK and -DSAEK bilateral corneal allograft rejection after vaccination were reported. These cases suggest a systemic inflammatory etiology for corneal graft rejection [33,38]. The detailed pathophysiology of the relationship between COVID-19 vaccination and corneal allograft rejection remains unclear. However, the present study revealed cases with minimal confounding variables and appreciable temporal correlation with COVID-19 vaccination, positing the vaccine’s role in inducing acute corneal allograft rejection.

Corneal transplantation is associated with low rates of allograft rejection, probably due to ocular immune privilege [52,53,54]. This underscores the importance of recognizing the possibility of COVID-19 vaccine-led corneal allograft rejection in the 21 patients (23 eyes) included in the present study [20,29,30,31,32,33,34,35,36,37,38,39,40]. The interval between administration of the COVID-19 vaccine and corneal graft rejection ranged from 1 day to 6 weeks (mean: 10.4 days; median: 7 days [20,29,30,31,32,33,34,35,36,37,38,39,40].) The concerns regarding vaccine-associated acute allograft rejection extend beyond the COVID-19 vaccines, especially those related to influenza, hepatitis B, tetanus, and yellow fever viral vaccines [21,55,56]. The incidence rate of vaccine-associated corneal graft rejection is certainly modest in terms of corneal transplant frequency. However, the projected societal shift towards a more frequent vaccination schedule calls for clinicians to be cognizant of a possible connection between the temporality of vaccine administration and graft rejections. Cell-mediated immune responses were confirmed in previous studies for the vaccine types included in this systematic review, including BNT162b2 (Pfizer), mRNA-1273 (Moderna), ChAdOx1 (AstraZeneca), and CoronaVac (Sinovac) vaccines [57,58,59,60,61,62]. Regardless of the vaccine type, vaccination significantly increased anti-spike-neutralizing antibodies, antigen-specific CD4^+^ T-cell responses, and inflammatory cytokines, including interferon (IFN)-γ and interleukin-2 [57,58,59,60,61,62]. IFN-γ plays a central role in the acute rejection process [63], and the resultant T helper type 1–dominant immune response may have evoked corneal allograft rejection in the vaccinated individuals [63].

Another risk factor for vaccine-associated corneal allograft rejection may be the presence of a corneal bed with a high rejection risk. Recurrent infections, autoimmune disease complications, and multiple corneal transplantations lead to progressive neovascularization and lymphangiogenesis of the cornea, virtually eliminating the immune privilege of the anterior eye [64]. Such corneal beds have a 40–90% rejection rate in subsequent transplantations. Of the 23 eyes included in this review, nine had undergone more than one transplantation [29,31,35,36,37,39,40]. Rallis et al. reported a case of an acute corneal allograft rejection following BNT162b2 vaccine administration with a surgical history of DSAEK and a re-do PKP for an existing Fuchs’ corneal endothelial dystrophy [31]. This suggests that the angiogenesis and lymphangiogenesis induced by repeated insults to the cornea (e.g., repeated corneal allograft transplantation) may predispose the corneal bed to the high immune stress that follows COVID-19 vaccine administration. Consequently, high-risk allograft recipients who have undergone repeated corneal transplants should be monitored and thoroughly examined after vaccination. Nine of the eyes included in this review developed corneal graft failure after vaccination [29,31,35,36,37,39,40], all of which had undergone repeated corneal transplants. As inadequate control of the corneal immune activity may subject the allograft to inflammatory insult, even from minor vaccinations, owing to its newly developed systemic communication, high-risk allograft recipients should be frequently followed up with appropriate immune-suppressive management.

Interestingly, acute cases of graft rejection of organs other than the cornea after the administration of the COVID-19 vaccines have been relatively rare, with only one case reported to date [65]. Excluding one patient (one eye) from the total reviewed patients [34], none of the patients received oral steroids or immune-modulatory medications. This contrasts with other organ transplant cases, where lifelong immunosuppressive or steroid therapy is typically prescribed [66]. Furthermore, the dampened inflammatory responses to the vaccine may have reduced the immune stress on these grafted organs. Therefore, prescribing oral immunomodulators or increasing the frequency of topical steroid administration should be considered for allograft recipients with a high rejection risk because of continued angiogenesis or lymphangiogenesis, particularly from 7 to 28 days after COVID-19 vaccine administration, when immune responses are at their peak [58].

This systematic review has certain limitations. First, the number of reported cases reviewed in this study is limited because of the recency of the ongoing pandemic. COVID-19 vaccines were approved and made clinically available in December 2020; therefore, only 21 cases with corneal allograft rejection after receiving the COVID-19 vaccine have been reported. Consequently, this study did not conduct a meta-analysis for the specific outcomes. Second, the generalizability of the results should be considered with caution because the studies included are from eight different countries with an ethnically white-predominant subject pool. There are no reports on patients from East Asia, and future investigations should aim to accrue cases from various geographical and ethnic backgrounds. Finally, the results of this systematic review do not elucidate the detailed pathophysiology of acute allograft rejection. Although two of the included studies performed anterior chamber aqueous humor PCR [20,38], the remaining studies did not successfully rule out any confounding causes of corneal graft rejection. In addition, this study included patients who experienced corneal graft failure >3 weeks after vaccination [40]. The COVID-19 vaccination may not have had direct links to these reported cases of corneal graft rejection. Therefore, the 21 cases of graft failure included in this review may have confounding aspects beyond the direct effects of the COVID-19 vaccine, including inadequate immune suppression at the time of vaccination. Future studies should incorporate extensive examinations—such as corneal endothelial cell density and viral panels—to remove the effects of confounders that may cause corneal graft failure and isolate vaccine-induced corneal changes. Nonetheless, the cases of two patients with a stable post-transplantation course over 20 years who developed signs of rejection after receiving the COVID-19 vaccine [33,38] and the bilaterality of some included cases suggest that the systemic inflammation and immune system upregulation caused by the COVID-19 vaccine may be associated with acute corneal allograft rejection [33,38].

In conclusion, this systematic review identified clinical features and host factors associated with corneal allograft rejection after COVID-19 vaccination. As the virus continues to spread, additional booster COVID-19 vaccine schedules are expected. Therefore, proper follow-up of corneal allograft recipients and interventions to prevent corneal allograft rejection after administering the COVID-19 vaccine may be crucial.

## Figures and Tables

**Figure 1 jcm-11-04500-f001:**
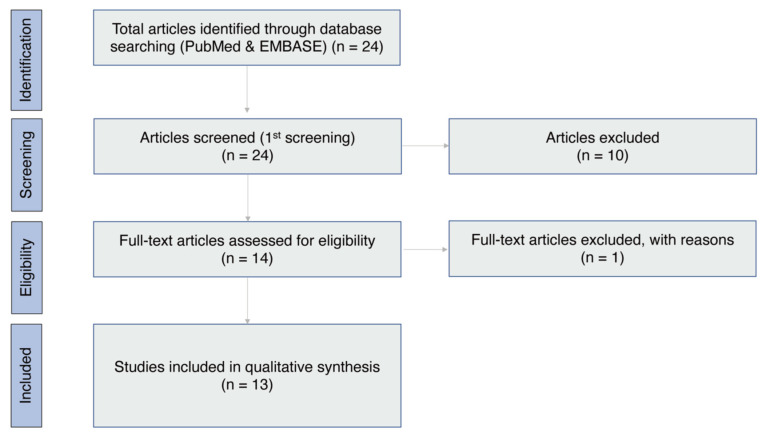
PRISMA flowchart illustrating the article selection process. PRISMA, Preferred Reporting Items for Systematic Reviews and Meta-Analyses.

**Table 1 jcm-11-04500-t001:** Inclusion and exclusion criteria.

Inclusion Criteria	Exclusion Criteria
1. Population: patients with corneal graft rejection after COVID-19 vaccination	1. Clinical guidelines, consensus documents, reviews, systematic reviews, and conference proceedings
2. Study design: retrospective studies (cross-sectional studies, case-control studies, case series, and case reports) and prospective studies	2. Articles on corneal graft rejection after SARS-CoV-2 infection
3. Outcomes: age, sex, and race of patients; type of vaccine; the number of vaccinations, type of corneal transplantation; the interval between corneal transplantation and rejection; ocular findings, including clinical ocular signs; post-vaccination period; medications for rejection, and corneal graft failure.	3. Articles on corneal graft rejection after the vaccinations for other viral infections
	4. Articles on animal-based studies
	5. Preprint articles
	6. Conference abstracts

Abbreviations: COVID-19, coronavirus disease 2019; SARS-CoV-2, severe acute respiratory syndrome coronavirus 2.

**Table 2 jcm-11-04500-t002:** Risk of bias assessment for included articles using the Joanna Briggs Institute Critical Appraisal Checklist for Case Reports and Case Series.

Source	Study Type	Question (Case Report)										Question (Case Series)											
		1	2	3	4	5	6	7	8	% Yes	Risk	1	2	3	4	5	6	7	8	9	10	% Yes	Risk
Phylactou et al. [20]	CS											Y	Y	Y	N	N	Y	Y	Y	U	NA	60	Moderate
Wasser et al. [29]	CS											Y	Y	Y	N	N	Y	Y	Y	Y	NA	70	Low
Crnej et al. [30]	CR	Y	Y	Y	Y	Y	Y	Y	Y	100	Low												
Rallis et al. [31]	CR	Y	Y	Y	Y	Y	Y	Y	Y	100	Low												
Abousy et al. [32]	CR	Y	Y	Y	Y	Y	Y	Y	Y	100	Low												
Shah et al. [33]	CS											Y	Y	Y	N	N	Y	Y	Y	U	NA	60	Moderate
de la Presa et al. [34]	CR	Y	Y	Y	Y	Y	Y	Y	Y	100	Low												
Yu et al. [35]	CR	Y	Y	Y	Y	Y	Y	Y	Y		Low												
Balidis et al. [36]	CS											Y	Y	Y	N	N	Y	Y	Y	U	NA	60	Moderate
Parmar et al. [37]	CR	Y	Y	Y	Y	Y	Y	Y	Y	100	Low												
Nioi et al. [38]	CR	Y	Y	Y	Y	Y	Y	Y	Y	100	Low												
Simão and Kwitko [39]	CR	Y	Y	Y	Y	Y	Y	Y	Y	100	Low												
Rajagopal and Priyanka [40]	CR	Y	Y	Y	Y	Y	Y	Y	Y	100	Low												

Abbreviations: CS, case series; CR, case report; Y, yes; N, no; U, unclear; NA, not applicable.

**Table 3 jcm-11-04500-t003:** Characteristics of the included articles.

Source	Publication Date	Study Type	Country	COVID-19 Vaccination		Corneal Transplantation									
				Number	Age (y)/Sex	Race	Type of Vaccine	Number of Vaccine Doses	Type of Corneal Transplantation	Interval between Corneal Transplantation and Rejection	Interval between Vaccination and Rejection	Ocular Findings	Medication (s)	Graft Failure	Other Findings
Phylactou et al. [20]	29 April 2021	CS	United Kingdom	2	66/F	Caucasian	BNT162b2	2	DMEK	14 d	17 d	Circumcorneal injection, KPs, and AC inflammation	Dexamethasone 0.1% eye drops hourly	No	Medical history of well-controlled HIV infection. Negative PCR test results with primers for cytomegalovirus, herpes simplex virus, and varicella-zoster virus.
					83/F	Caucasian	BNT162b2	1	DMEK (OD)	6 y	3 wk	Circumcorneal injection, KPs, and AC inflammation	Dexamethasone 0.1% eye drops hourly	No	Bilateral, simultaneous acute endothelial graft rejection.
									DMEK (OS)	3 y	3 wk	ciliary injection, epithelial and stromal edema, fluid at the LASIK interface, and AC reaction	Dexamethasone 0.1% eye drops hourly	No	
Wasser et al. [29]	24 May 2021	CS	Israel	2	73/M	NA	BNT162b2	1	PKP	2 y	13 d	Diffuse corneal edema, KPs, and AC cells	Dexamethasone 0.1% eye drops hourly and oral prednisone 60 mg daily	Yes	A PKP reoperation case with allograft rejection after vaccination. Before vaccination, the patient was on 0.1% dexamethasone eye drops once daily.
					56/M	NA	BNT162b2	1	PKP	10 mo	14 d	Graft edema with fine endothelial KPs	Dexamethasone 0.1% eye drops hourly and oral prednisone 60 mg daily	Yes	A PKP reoperation case with allograft rejection after vaccination.
Crnej et al. [30]	20 July 2021	CR	Lebanon	1	71/M	NA	BNT162b2	1	DMEK	5 mo	7 d	Ciliary injection, diffuse corneal edema within the graft, KPs, DF, and AC cells	Dexamethasone sodium phosphate 1 mg/mL eye drops and oral valacyclovir 1000 mg three times daily	No	The second vaccine was administered while continuing steroid eye drops, and no signs of graft rejection were noted after the vaccine administration
Rallis et al. [31]	24 August 2021	CR	United Kingdom	1	68/M	NA	BNT162b2	1	PKP	4 mo	3 d	Conjunctival hyperemia and epithelial rejection line	Dexamethasone 0.1% eye drops hourly and oral acyclovir 400 mg five times daily for 1 week	Yes	The left eye with allograft rejection underwent repeated PKP after DSAEK. However, the right eye, which underwent only DSAEK, showed no evidence of graft rejection.
Abousy et al. [32]	13 September 2021	CR	America	1	73/M	NA	BNT162b2	2	DSAEK (OD)	8 y	4 d	Thickened corneas with DF	Prednisolone acetate 1% eye drops four times daily	No	Each eye of the patient received a graft from a different donor.
									DSAEK (OS)	8 y	9 d	Moderate conjunctival congestion, diffuse corneal edema, KPs, and AC inflammation	Prednisolone acetate 1% eye drops four times daily	No	
Shah et al. [33]	8 October 2021	CS	America	4	74/M	Caucasian	mRNA-1273	1	DMEK	5 mo	1 wk	Conjunctival inflammation, corneal endothelial rejection line with KPs, and diffuse edema	Prednisolone acetate 1% eye drops every 2 h	No	Before vaccination, the patient was on 0.1% fluorometholone ophthalmic drops.
					61/F	Caucasian	mRNA-1273	2	PKP	3 y	1 wk	Conjunctival injection, AC cells, and corneal stromal edema	The frequency of prednisolone acetate 1% eye drops was increased to hourly.	No	Before vaccination, the patient was on 1% prednisone eye drops once daily.
					69/F	Black	mRNA-1273	2	DSAEK	6 y	2 wk	Mild conjunctival hyperemia and injection, mild corneal edema, AC flare and cells, and KPs	Difluprednate 0.05% eye drops six times daily	No	Before vaccination, the patient was on 1% prednisone eye drops once daily. The patient had undergone DSAEK in both eyes, but graft rejection occurred only in the left eye.
					77/M	Caucasian	mRNA-1273	2	PKP	22 y	1 wk	Subtle corneal edema and small pigmented KPs	Prednisolone acetate 1% eye drops five times daily	No	No history of pre-vaccination eye drop administration.
de la Presa et al. [34]	7 October 2021	CR	America	1	27/F	NA	mRNA-1273	1	LR-CLAL	4 y	15 d	Conjunctival injection and Diffuse corneal edema	Difluprednate 0.05% eye drops every hour, oral prednisone 30 mg daily, and oral mycophenolate mofetil 500 mg twice daily	No	Before vaccination, the patient was under treatment with mycophenolate mofetil 500 mg twice daily orally and prednisolone 1% eye drops twice daily. The patient was administered the second dose of vaccine during the escalation of the oral drug dose and did not develop graft rejection.
Yu et al. [35]	25 October 2021	CR	America	1	51/M	NA	mRNA-1273	1	PKP	3 wk	3 d	Conjunctival hyperemia, corneal graft edema, DF, KPs, and AC activity	The frequency of topical steroid eye drops was increased to every 2 h.	Yes	This was a PKP reoperation case with allograft rejection after vaccination. In addition, the patient had a history of steroid-induced glaucoma. Postoperatively, both antibiotic and steroid eye drops were administered four times daily.
Balidis et al. [36]	26 November 2021	CS	Greece	4	77/F	Caucasian	mRNA-1273	1	DMEK	20 mo	1 wk	Diffuse corneal edema and inflammation in the AC	Subconjunctival dexamethasone injections and eye drops of 1 mg/mL dexamethasone and hypertonic every 2 h	No	Previously, the patients had undergone a total of two DMEK procedures.
					64/F	Caucasian	mRNA-1273	2	PKP	2 y	1 wk	Diffuse corneal edema and KPs	Dexamethasone eye drops hourly and intracameral fortecortin injections	Yes	The patient had undergone repeated PKP procedures.
					69/M	Caucasian	ChAdOx1	1	PKP	2 y	5 d	Corneal edema	Subconjunctival dexamethasone injections and combined oral (methylprednisolone) and topical (dexamethasone) corticosteroid therapy	No	The patient had herpetic keratitis and was taking oral valacyclovir.
					63/M	Caucasian	ChAdOx1	1	DSAEK	1 y	10 d	Stromal edema suggestive of a stromal rejection	Dexamethasone eye drops and hypertonic ointment every 2 h	Yes	Previously, the patients had undergone a total of two DSAEK procedures.
Parmar et al. [37]	27 November 2021	CR	India	1	35/M	NA	ChAdOx1	1	PKP	6 mo	2 d	Ciliary injection, corneal edema, DF, and KPs	Atropine sulfate 1% eye drops hourly and intravenous methyl prednisone 1000 mg once a day over 3 h for 3 days	Yes	The patient was relatively young and had undergone re-do PKP for a previous failed large, vascularized therapeutic graft.
Nioi et al. [38]	27 November 2021	CR	Italy	1	44/F	Caucasian	BNT162b2	1	PKP	25 y	13 d	Thickened cornea with DF	Dexamethasone 0.2% eye drops hourly and 1000 IU of Vitamin D supplement (cholecalciferol, DIBASE^®^) daily	No	PCR test results for herpes simplex and other viruses were negative, but blood tests showed severe vitamin D deficiency.
Simão and Kwitko [39]	15 December 2021	CR	Brazil	1	63/F	NA	CoronaVac	1	PKP	7 y	1 d	Corneal edema, endothelial rejection line, epithelial and stromal edema, DF, and KPs	Dexamethasone eye drops hourly, and 0.5% timolol maleate and 0.03% bimatoprost polydimethylsiloxane eye drops four times a day	Yes	The patient had undergone three previous PKP procedures.
Rajagopal and Priyanka [40]	23 December 2021	CR	India	1	79/M	NA	ChAdOx1	2	PKP	4 y	6 wk	Microcystic epithelial and stromal corneal graft edema and endothelial KPs	Steroid eye drops hourly and oral steroids	Yes	The patient did not have pain or photophobia but had a history of repeated transplant surgeries.

Abbreviations: CS, case series; CR, case report; F, female; DMEK, Descemet membrane endothelial keratoplasty; d, days; KP, keratic precipitates; AC, anterior chamber; HIV, human immunodeficiency virus; PCR, polymerase chain reaction; OD, oculus dextrus; wk, weeks; OS, oculus sinister; LASIK, laser-assisted in situ keratomileusis; M, male; NA, not applicable; PKP, penetrating keratoplasty; y, years; mo, months; DF, Descemet’s folds; DSAEK, Descemet stripping automated endothelial keratoplasty; LR-CLAL, living-related conjunctival-limbal allograft.

**Table 4 jcm-11-04500-t004:** Clinical ocular manifestations after COVID-19 vaccination.

Clinical Ocular Manifestations	Number of Eyes (*n* = 23)	Percentage (%)
Corneal edema	20	87.0
Keratic precipitates	14	60.9
Conjunctival or ciliary injection	14	60.9
Inflammation reaction in the anterior chamber	10	43.5
Descemet membrane folds	6	26.1
Corneal endothelial rejection line	3	13.0
Fluid at the LASIK interface	1	4.3

Abbreviations: COVID-19, coronavirus disease 2019; LASIK, laser in situ keratomileusis.

**Table 5 jcm-11-04500-t005:** Medications used for corneal graft rejection.

Medications	Number of Eyes (*n* = 23)	Percentage (%)
Topical corticosteroids only	12	52.1
Topical and oral corticosteroids	4	13
Topical corticosteroids and subconjunctival or intracameral corticosteroid injections	2	8.7
Topical and intravenous corticosteroids	1	4.3
Topical and oral corticosteroids and subconjunctival corticosteroid injections	1	4.3
Topical and oral corticosteroids and immunosuppressants	1	4.3
Topical corticosteroids and vitamin D supplements	1	4.3

## Data Availability

All data generated or analyzed during this study are included in this published article.

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
