# Peer review of "Characteristics and Clinical Ocular Manifestations in Patients with Acute Corneal Graft Rejection after Receiving the COVID-19 Vaccine: A Systematic Review"

_jcm, 2022, doi:10.3390/jcm11154500_

Round 1
Reviewer 1 Report
This is a highly relevant topic. The literature research was performed properly. In my opinion the statistical analysis should be using the guidelines of a Cochrane review.
This includes
- a much more specific ranking and evaluation of the included literature (scoring system regarding different fields of bias)
- include forest plots for the different outcomes
Author Response
Responses to comments from Reviewer 1:
- This is a highly relevant topic. The literature research was performed properly. In my opinion the statistical analysis should be using the guidelines of a Cochrane review.
[Response]
We want to thank the reviewer for evaluating our manuscript. We conducted our systematic review according to Cochrane review guidelines. In addition, we have added the citation below in the revised manuscript.
Lines 144–146
Data analyses were performed considering the Updated Method Guidelines for Systematic Reviews in the Cochrane Collaboration Back Review Group [24].
- This includes
- a much more specific ranking and evaluation of the included literature (scoring system regarding different fields of bias)
[Response]
Thank you for your suggestion. We have added the risk of bias assessment for the included literature in the new Table 2 and the Methods section, as shown below.
Lines 120–130
Risk of bias assessment
The risk of bias in the individual studies was assessed using either the Joanna Briggs Institute (JBI) Critical Appraisal Checklist for Case Reports or the JBI Critical Appraisal Checklist for Case Series [22]. The checklists for case reports and case series consist of 8 and 10 items, respectively, with response choices of “yes,” “no,” “unclear,” or “not applicable.” Two investigators (K.F. and K.N.) independently assigned an overall risk of bias to each eligible study, and if they disagreed, a third reviewer (T.I.) was consulted. The risk of bias was determined considering the total number of “yes” responses, with ≥70%, 50–69%, and ≥49% of the responses indicating low, moderate, and high risk of bias, respectively [23].
Lines 159–162
The results of the JBI Critical Appraisal Checklists for case reports and case series are summarized in Table 2. The 13 articles showed a low or moderate risk of bias.
- – include forest plots for the different outcomes
[Response]
Thank you for this comment. Unfortunately, we conducted the systematic review without a meta-analysis because reports of corneal graft rejection after COVID-19 vaccine administration are case reports and case series. Therefore, we could not conduct a meta-analysis using forest plots. We have described this as a limitation, as shown below.
Lines 335–341
This systematic review has certain limitations. First, the number of reported cases reviewed in this study is limited because of the recency of the ongoing pandemic. COVID-19 vaccines were approved and made clinically available in December 2020; therefore, only 21 cases with corneal allograft rejection after receiving the COVID-19 vaccine have been reported. Consequently, this study did not conduct a meta-analysis for the specific outcomes.

Reviewer 2 Report
Thank you very much for giving me the opportunity to review this work entitled "Characteristics and clinical ocular manifestations in pa-2 tients with acute corneal graft rejection after receiving 3 the COVID-19 vaccine: a systematic review.”
The authors present a very well written article with an interesting idea of research. Tables are very good tables, all the information are there, even they present time between surgery and rejection AND between vaccine and rejection. In Line 87. Authors present idea about management and consequences. Regarding that there was even some case of rejection after 14 days (Phylactou et al [16].) and I guess this patient had an intensive treatment before rejection (4-6 times maxidex depend of surgeon’s preferences) This makes me think and share the same idea that our colleagues. Then congratulations for your discussion.
The introduction is very well writing and very well discussed. Material and methods were correct. However, I consider that some inputs should be taken in count.
1. Line 64 and introduction: (“The concerns for vaccine-associated 64 acute allograft rejection extend beyond the COVID-19 vaccines, especially related to in-65 fluenza, hepatitis B, tetanus, and yellow fever viral vaccines [47,48].”)
In this line they talked about previous rejection with other vaccines, I strongly believe that this idea should be introduce in the introduction too. Even if they can give more information about if it is most common with one specifically will be great. I understand that they don’t need to do an extensive review about “other vaccines and rejection” but would be great understand if there is a big background history of rejection with other vaccines or only with this. Actually, they should happen for the same concept (inmunorejection)
In summary, the article is well written and authors exposes their ideas and limitation.
Author Response
Responses to comments from Reviewer 2:
1.- The authors present a very well written article with an interesting idea of research. Tables are very good tables, all the information are there, even they present time between surgery and rejection AND between vaccine and rejection. In Line 87. Authors present idea about management and consequences. Regarding that there was even some case of rejection after 14 days (Phylactou et al [16].) and I guess this patient had an intensive treatment before rejection (4-6 times maxidex depend of surgeon’s preferences) This makes me think and share the same idea that our colleagues. Then congratulations for your discussion.
[Response]
Thank you for the constructive critique to improve our manuscript. We have made every effort to address the issues raised and to respond to all comments.
- Line 64 and introduction: (“The concerns for vaccine-associated acute allograft rejection extend beyond the COVID-19 vaccines, especially related to influenza, hepatitis B, tetanus, and yellow fever viral vaccines [47,48].”)
[Response]
Accordingly, we have expounded on other vaccine-associated acute allograft rejections in the Introduction, as shown below:
Lines 75–78
In addition, several reports have previously described corneal allograft rejection triggered by vaccinations, including those secondary to Influenza and Hepatitis B vaccine administration [17].

Reviewer 3 Report
- Congratulations to the authors for this article.
- The manuscript is appropriately referenced and authors presented sufficient data with appropriate tables and figures and the article is easy to read and logically structured.
- The review reported only 21 cases of graft rejection after vaccination but I think thousands of cases with corneal grafts were vaccinated without rejection, so another factor may have a role.
- In Page 3 ; search strategy "An extensive search strategy was designed to retrieve all articles published from April 29, 2021, to December 23, 2021” while in page 4 in section 3.1 “The articles included in this systematic review were published between January 15, 2021, and December 23, 2021 ”. Please correct.
- The pathology (Indication for corneal transplantation) was not mentioned.
- The protocol for treatment of graft rejection was not the same in all cases. So graft rejection may be related to vaccine but graft failure may be due to vaccine or inappropriate management of graft rejection.
Author Response
Responses to comments from Reviewer 3:
- The manuscript is appropriately referenced and authors presented sufficient data with appropriate tables and figures and the article is easy to read and logically structured.
[Response]
Thank you for your reviewing our manuscript. We have revised our manuscript according to your suggestions.
- The review reported only 21 cases of graft rejection after vaccination but I think thousands of cases with corneal grafts were vaccinated without rejection, so another factor may have a role.
[Response]
We agree with your comment that vaccine-associated corneal graft rejection is uncommon, given the total number of corneal transplants performed. Therefore, we have added this information to the manuscript, as shown below. In addition, the pathology of vaccine-associated corneal graft rejection remains unclear. Hence, we have described this as a limitation in the Discussion section (Lines 280–282, 346–350).
Lines 284–288
The incidence rate of vaccine-associated corneal graft rejection is certainly modest in terms of corneal transplant frequency. However, the projected societal shift towards a more frequent vaccination schedule calls for clinicians to be cognizant of a possible connection between the temporality of vaccine administration and graft rejections.
Lines 345–346
Finally, the results of this systematic review do not elucidate the detailed pathophysiology of acute allograft rejection.
- In Page 3 ; search strategy "An extensive search strategy was designed to retrieve all articles published from April 29, 2021, to December 23, 2021” while in page 4 in section 3.1 “The articles included in this systematic review were published between January 15, 2021, and December 23, 2021 ”. Please correct.
[Responses]
Lines 103
Thank you for this comment. We have revised the text from “An extensive search strategy was designed to retrieve all articles published from April 29, 2021, to December 23, 2021” to “An extensive search strategy was designed to retrieve all articles published until February 23, 2022.”
- The pathology (Indication for corneal transplantation) was not mentioned.
[Responses]
Thank you for this suggestion. We have included the ocular findings for the diagnosis of corneal rejection in Table 3. However, each article has different diagnostic criteria for corneal rejection; consequently, we have added this as a limitation in the revised manuscript. (Lines 352–358)
Lines 346–348
Finally, the results of this systematic review do not elucidate the detailed pathophysiology of acute allograft rejection.
- The protocol for treatment of graft rejection was not the same in all cases. So graft rejection may be related to vaccine but graft failure may be due to vaccine or inappropriate management of graft rejection.
[Responses]
We agree with your comments; hence, we have added the sentences below to the revised manuscript.
Lines 315–320
Nine of the eyes included in this review developed corneal graft failure after vaccination [25,27,31-33,35,36], all of which had undergone repeated corneal transplants. As inadequate control of the corneal immune activity may subject the allograft to inflammatory insult, even from minor vaccinations, owing to its newly-developed systemic communication, high-risk allograft recipients should be frequently followed up with appropriate immune-suppressive management.
Lines 350–356
In addition, this study included patients who experienced corneal graft failure >3 weeks after vaccination [36]. The COVID-19 vaccination may not have had direct links to these reported cases of corneal graft rejection. Therefore, the 21 cases of graft failure included in this review may have confounding aspects beyond the direct effects of the COVID-19 vaccine, including inadequate immune suppression at the time of vaccination.
